# CDK4/6 Inhibitor Resistance in Hormone Receptor-Positive Metastatic Breast Cancer: Translational Research, Clinical Trials, and Future Directions

**DOI:** 10.3390/ijms241411791

**Published:** 2023-07-22

**Authors:** Jin Sun Lee, Hannah Hackbart, Xiaojiang Cui, Yuan Yuan

**Affiliations:** 1Department of Medicine, Samuel Oschin Comprehensive Cancer Institute, Cedars-Sinai Medical Center, Los Angeles, CA 90048, USA; 2Department of Surgery, Samuel Oschin Comprehensive Cancer Institute, Cedars-Sinai Medical Center, Los Angeles, CA 90048, USA

**Keywords:** breast cancer, CDK4/6 inhibitors, estrogen receptor, therapeutic resistance

## Abstract

The emergence of CDK4/6 inhibitors, such as palbociclib, ribociclib, and abemaciclib, has revolutionized the treatment landscape for hormone receptor-positive breast cancer. These agents have demonstrated significant clinical benefits in terms of both progression-free survival and overall survival. However, resistance to CDK4/6 inhibitors remains a challenge, limiting their long-term efficacy. Understanding the complex mechanisms driving resistance is crucial for the development of novel therapeutic strategies and the improvement of patient outcomes. Translational research efforts, such as preclinical models and biomarker studies, offer valuable insight into resistance mechanisms and may guide the identification of novel combination therapies. This review paper aims to outline the reported mechanisms underlying CDK4/6 inhibitor resistance, drawing insights from both clinical data and translational research in order to help direct the future of treatment for hormone receptor-positive metastatic breast cancer.

## 1. Introduction

Hormone receptor (HR)-positive metastatic breast cancer (MBC) is characterized by the expression of an HR (estrogen receptor and/or progesterone receptor) and the absence of human epidermal growth factor receptor-2 (HER2) overexpression. Although endocrine-based therapy has improved the prognosis of this subtype, HR-positive MBC (HR+ MBC) continues to present a significant clinical challenge and is responsible for a substantial number of breast cancer deaths [1]. However, the emergence of cyclin-dependent kinase 4 and 6 (CDK4/6) inhibitors, including palbociclib, ribociclib, and abemaciclib, has transformed the treatment landscape for HR+ MBC. When combined with endocrine therapy, these agents have demonstrated remarkable efficacy, leading to improved progression-free survival (PFS) and overall survival (OS) rates, as shown in Table 1. Building on the success of CDK4/6 inhibitors in the metastatic setting, further studies have been conducted to assess their efficacy in early breast cancer. For instance, abemaciclib was recently approved for use in the adjuvant setting based on the results of the MonarchE trial, which revealed sustained benefits in invasive disease-free survival (iDFS) even after the discontinuation of the drug [2]. Furthermore, ribociclib has also demonstrated improved iDFS in patients with stage II or III high-risk HR+ breast cancer when used for 3 years in the adjuvant setting [3].

CDK4/6 inhibitors exert their function through the targeted inhibition of kinases CDK4/6, which play crucial roles in regulating cell cycle progression and proliferation [17], particularly the transition from the G1 phase to the S phase. Upon binding with cyclin D, CDK 4/6 form an active complex that triggers DNA synthesis and entry into the S phase via the phosphorylation of the tumor suppressor protein retinoblastoma (Rb). This phosphorylation event releases the repression of the E2F family, facilitating the hyperphosphorylation of Rb via the activities of cyclin E1 and E2. This cascade further promotes the expression of E2F target genes that are crucial for cell progression into the S phase (Figure 1). By inhibiting these kinases, CDK 4/6 inhibitors induce cell cycle arrest in the G1 phase, thereby preventing entry into the S phase and subsequent DNA synthesis. This mechanism of action effectively disrupts tumor growth and slows down disease progression. It is of note that CDK4/6 have also been demonstrated to phosphorylate proteins involved in cell differentiation, apoptosis, and mitochondrial function. This may partially explain the broad anti-tumor effects of CKD 4/6 inhibitors [18]. All three CDK4/6 inhibitors used in the clinical setting demonstrate the inhibition of both CDK 4 and CDK 6. However, abemaciclib and ribociclib specifically exhibit greater potency against CDK4 compared to CDK6. Additionally, abemaciclib inhibits multiple other kinases, including CDK1 and CDK2 [19]. These differences in potency may lead to varying side effects. The most commonly observed adverse events for palbociclib and ribociclib are cytopenia, such as neutropenia and anemia, while abemaciclib is associated with diarrhea. Additionally, a rare unique adverse event of QT prolongation has been reported with ribociclib. With the wide range of effects of kinases on the progression of the cell cycle, these drugs may have multiple mechanisms through which they contribute to cancer cell death.

Despite the initial clinical success of CDK 4/6 inhibitors, the development of resistance remains a critical challenge in the long-term management of HR+ MBC. Unfortunately, many patients will continue to experience disease progression despite receiving continuous treatment with these agents. Additionally, the recent approval of abemaciclib in the adjuvant setting will certainly present new clinical challenges in determining treatment strategies for patients experiencing relapse while on or following the use of CDK 4/6 inhibitors. Understanding the underlying mechanisms of resistance is essential in developing strategies to overcome this limitation and improve patient outcomes. Further research and explorations of novel combinations are needed to address these challenges and advance treatment options for resistant disease.

This review aims to comprehensively explore the mechanisms of CDK 4/6 inhibitor resistance, drawing insights from both clinical data and translational research. By examining the available evidence, we seek to identify the molecular alterations, signaling pathways, and cellular processes that contribute to resistance development. Furthermore, we will highlight the implications of these resistance mechanisms on clinical practice and discuss potential strategies to overcome these resistance patterns in order to enhance treatment efficacy.

## 2. Potential Mechanisms of CDK 4/6 Inhibitor Resistance in Preclinical Studies

### 2.1. Alterations in Cell Cycle Pathway Components

The CDK 4/6 pathway plays a crucial role in cell cycle progression and is the primary target of CDK 4/6 inhibitors. Alterations in various components of this pathway can contribute to CDK 4/6 inhibitor resistance. One potential mechanism involves alterations to cyclin D1, the primary activator of CDK 4/6. For example, the amplification or overexpression of cyclin D1 have been observed in a subset of patients who develop resistance to CDK 4/6 inhibitors [20]. In contrast, another study demonstrated that mutations activating cyclin D could potentially increase sensitivity to CDK4/6 inhibitors. Meanwhile, a deficiency in cyclin D, which may be attributed to the oncogene addiction property in some cancer cells, is associated with resistance to CKD4/6 inhibitors [21,22]. Mutations or alterations in kinases CDK4 and 6 themselves have also been implicated in causing resistance [23]. These alterations can impair the binding of CDK 4/6 inhibitors to their targets, reducing the inhibitory effect and diminishing the response to therapy. For example, mutations in the ATP-binding pocket of CDK4/6 may interfere with the binding of CDK 4/6 inhibitors, thereby conferring resistance to treatment [24]. 

The relationships between tumor cell cycle mediators are highly heterogenous, influenced by tumor origin and genetic alterations. These overlapping pathways allow for the development of CDK4/6 resistance via the utilization of bypass mechanisms. Knudsen et al. demonstrated diverse patterns of dependency of CDK4/6 in breast cancer cells for cell cycle progression [25]. While some cells conform to a conventional CDK4/6-dependent cycle, others can bypass this dependency through mechanisms involving cyclin E activation or cyclin D1. Tumor cells with Rb loss display heterogeneous CDK and cyclin requirements, with some relying on cyclin E for progression. Moreover, Palafox et al. showed that p16 overexpression was associated with reduced response to CDK4/6 inhibitors in patient-derived xenografts and breast cancer cell lines, while heterozygous *Rb1* loss served as a biomarker for acquired resistance and poor clinical outcomes [26]. Cyclin E2 (*CCNE2*) overexpression was also proposed as a potential mechanism of resistance to both endocrine therapy and CDK4 inhibition from preclinical studies [27], and *CCNE1* overexpression was associated with a lower response to palbociclib in the PALOMA-3 trial [28]. Taken together, these findings highlight the complex nature of cancer cell cycles, including multiple mechanisms for G1/S progression and differential dependencies on CDKs and cyclins. Modifications or mutations across various elements of these pathways can lead to resistance against CDK4/6 inhibitors. It will be important to adopt precision therapeutic strategies that are tailored based on specific gene expressions and susceptibilities in order to overcome these resistance mechanisms in a clinically relevant way.

### 2.2. Activation of Alternative Pathways

In addition to alterations in major cell cycle regulators, CDK4/6 inhibitor resistance in breast cancer also involves various genomic alterations, with the RAS/MAPK and PI3K/AKT/mTOR pathways playing key roles. The PI3K/AKT/mTOR pathway is a critical signaling pathway involved in cell survival and proliferation. The activation of this pathway, often through genetic alterations or aberrant signaling, can promote resistance to CDK 4/6 inhibitors. The dysregulation of the PI3K/AKT/mTOR pathway can bypass the inhibitory effects of CDK 4/6 inhibitors and promote cell cycle progression, leading to disease progression despite treatment. O’Brien et al. demonstrated that treatment with a p110α-selective PI3K inhibitor, alpelisib (BYL719), completely blocked the progression of acquired CDK4/6 inhibitor-resistant xenografts, proposing a possible clinical solution for PI3K-driven resistant tumors [29]. Of note, a major cause of anti-cancer therapeutic resistance is the increase in multidrug resistance (MDR) genes, which are well-established targets of PI3K signaling [30]. A recent study found the upregulation of MDR proteins in CDK4/6 inhibitor-resistant breast cancer cells [31]. Based on these data, it is postulated that the PI3K/MDR axis may play a critical role in dictating the response to CDK4/6 inhibitors in MBC.

The loss of or alterations to neurofibromin 1 (*NF1*), a gene that regulates RAS and cell proliferation, have been linked to endocrine therapy resistance and disease progression, but its relationship with CDK4/6 inhibitors is still not well understood. One study suggested that patients with pathogenic *NF1* mutations may be more likely to develop resistance to CDK4/6 inhibitors. However, the impact of other *NF1* mutations and the potential for targeting the MAPK pathway in this context require further exploration [32].

The fibroblast growth factor receptor (FGFR) signaling pathway, which is known to be a critical regulator of breast cancer progression, has emerged as another potential mechanism of resistance to CDK 4/6 inhibitors [33,34]. The activation of FGFR signaling can drive cell proliferation and survival independent of CDK 4/6 activity. The activation of this pathway may confer resistance to CDK 4/6 inhibitors and limit their effectiveness.

### 2.3. Epigenetic Regulation and Transcriptional Rewiring

Epigenetic changes, such as chromatin remodeling and histone modifications, can also contribute to the development of resistance to CDK 4/6 inhibitors [35,36]. Chromatin remodeling involves altering the physical structure of DNA, affecting its access to various regulatory proteins [37,38]. These changes in structure can affect the binding of transcription factors to their target genes, leading to dysregulated gene expression patterns. Histone modifications, including acetylation, methylation, and phosphorylation, can also influence chromatin structure and gene transcription, causing the activation of resistance mechanisms. For example, histone deacetylases can increase the inhibition efficacy of CKD4/6 and mediate cell cycle arrest by upregulating p21 expression in CKD4/6 inhibitor-resistant tumors [39]. These various epigenetic modifications can modulate the expression of genes involved in the CDK 4/6 pathway and other resistance-associated pathways, ultimately contributing to further drug resistance development.

Transcription factors also play a crucial role in regulating gene expression. They function by binding to specific DNA sequences and modulating transcriptional activity. In the context of CDK 4/6 inhibitor resistance, the altered expression or activity of transcription factors can impact the response to therapy [40]. Transcription factors, such as E2F, AP-1, and NF-κB, can drive the expression of genes involved in cell cycle regulation, DNA repair, and cell survival, thereby promoting resistance. Additionally, enhancer hijacking, a phenomenon in which active enhancers are redirected to different genes, can lead to transcriptional rewiring and altered gene expression patterns [41]. This rewiring may result in the activation of pathways that bypass the inhibitory effects of CDK 4/6 inhibitors, conferring drug resistance. 

## 3. Clinical Trials and CDK 4/6 Inhibitor Resistance

Clinical manifestations of CDK 4/6 inhibitor resistance can vary greatly. A subset of patients with HR+ MBC demonstrate intrinsic or primary resistance to CDK 4/6 inhibitors, which presents as a lack of initial response or limited sustained benefit. Conversely, another subset of patients may initially exhibit a favorable response to CDK 4/6 inhibitor therapy but then develop disease progression despite continuous treatment, which is suggestive of acquired resistance. Careful clinical observation and the identification of resistance patterns can provide valuable insight into the essential characteristics of differing resistance patterns. Understanding the underlying factors that contribute to both intrinsic and acquired resistance is imperative in identifying patients who may not derive significant benefits from CDK 4/6 inhibitors, thus necessitating an exploration of alternative therapeutic strategies. Although extensive studies have focused on the identification of biomarkers that can predict poor responsiveness to CDK 4/6 inhibitors, the results have not been concordant. A further analysis of molecular alterations and genomic landscapes can help elucidate novel indicators that may predict a patient’s response to treatment. 

Turner et al. performed an analysis utilizing formalin-fixed paraffin-embedded (FFPE) tissue samples obtained from patients in the PALOMA-3 study to investigate the significance of varying gene expression levels on response to CDK 4/6 inhibitors [28]. In this study, the efficacy of palbociclib was lower in patients with elevated levels of expression of cyclin E1 (*CCNE1*) mRNA compared to those with lower expression levels. Notably, no significant interaction was identified between treatment efficacy and the expression levels of CDK4, CDK6, cyclin D1, and RB1. A separate investigation utilizing plasma samples from patients in the PALOMA-3 study revealed that early suppression of PIK3CA circulating tumor DNA (ctDNA) levels after 15 days of treatment with palbociclib and fulvestrant strongly correlated with improved PFS [42]. Furthermore, an additional investigation using paired baseline and end-of-treatment ctDNA sequencing from the PALOMA-3 study demonstrated that acquired mutations resulting from fulvestrant treatment, such as the *ESR1 Y537S* mutation, play a significant role in driving resistance to fulvestrant and palbociclib combination therapy [43].

Further biomarker studies have been conducted in different clinical trials. In the MONARCH-2 trial, a ctDNA analysis showed that treatment with abemaciclib plus fulvestrant improved PFS and OS in patients with PIK3CA and ESR1 mutations, as well as in those without these mutations, indicating the effectiveness of this combination therapy across certain mutation subgroups [44]. Similarly, a pooled ctDNA analysis from the MONALEESA phase III trial with 1503 patients identified specific genomic alterations, such as *FRS2* and *PRKCA*, that were associated with increased PFS from treatment with ribociclib, while alterations in genes such as *CHD4, BCL11B, ATM*, or *CDKN2A/2B/2C* provided limited benefit [45]. Another genomic study using human tissue samples identified losses of RB and PTEN expression as key drivers of acquired resistance to ribociclib and letrozole [46].

Wander et al. explored the genomic landscape of resistance to CDK4/6 inhibitors via the whole-exome sequencing (WES) of metastatic tumor tissues from patients who underwent treatment with CDK4/6 inhibitors [47]. This study revealed the heterogeneous genomic landscape of resistance mechanisms, identifying multiple potential mediators such as *Rb1* disruption, *AKT1* activation, and alterations in *AURKA, CCNE2, RAS, ERBB2*, and *FGFR2*. Lastly, our group conducted a comprehensive analysis of broad-panel next-generation sequencing (NGS) data obtained from patients receiving any CDK4/6 inhibitors in a real-world clinical setting [48]. This retrospective study analyzed genomic biomarkers of CDK4/6 inhibitor resistance in patients with HR-positive MBC treated with palbociclib, ribociclib, or abemaciclib. Genomic and RNA sequencing data were analyzed in relation to PFS, and associations were identified between specific genes. *FGFR1* amplification, *PTEN* loss, and DNA repair pathway gene mutations showed significant associations with shorter PFS in patients receiving treatment with CDK4/6 inhibitor therapy. These findings highlight the role potential genomic biomarkers could play in helping to predict response to CDK4/6 inhibitors in clinical practice.

In conclusion, clinical studies investigating treatment resistance to CDK4/6 inhibitors have provided valuable insights into this phenomenon. The emergence of resistance is characterized by clinical observations and patterns, with molecular alterations and the genomic landscape playing a crucial role. The findings from various studies investigating biomarkers have provided valuable insights, but the results have not always been consistent, leading to controversy and the need for further research.

## 4. Biomarkers for Predicting and Monitoring Resistance

Identifying biomarkers that can predict and monitor resistance to CDK4/6 inhibitors is crucial for optimizing treatment strategies and patient outcomes. Various approaches, such as genomic and transcriptomic profiling, ctDNA analysis, and imaging techniques, have been studied to identify and characterize biomarkers associated with CDK4/6 inhibitor resistance.

### 4.1. Tumor Tissue Profiling

The genomic and transcriptomic profiling of tumor samples have provided valuable insight into the molecular alterations associated with CDK 4/6 inhibitor resistance. For example, studies have identified alterations in genes such as *CCNE1, RB1, ERBB1*, and *ESR1* that contribute to resistance [28,46,48,49]. Genomic profiling using techniques like NGS has revealed the presence of specific mutations, amplifications, or deletions in these genes that can influence treatment response. Transcriptomic profiling has also uncovered gene expression signatures associated with CDK4/6 inhibitor resistance, providing potential predictive biomarkers. Recently, advanced single-cell sequencing has been used for genomic profiling. Griffiths et al. investigated the resistance pathways of CDK4/6 inhibitors and endocrine therapy using single-cell RNA sequencing (scRNAseq) [50]. The findings showed that combination therapy led to the emergence of resistance through a shift from estrogen signaling to the upregulation of JNK signaling proliferation through growth factor receptors. These types of analyses help to further dissect the mechanisms enabling resistance and introduce potential new targets for mitigating tumor growth.

### 4.2. Circulating Tumor DNA (ctDNA) Analysis

ctDNA analysis has emerged as a non-invasive approach for monitoring treatment response and detecting emerging drug resistance. Studies have demonstrated the utility of ctDNA analysis in identifying specific genetic alterations associated with CDK4/6 inhibitor resistance. For example, patients with a high ctDNA fraction and specific genomic alterations, such as *TP53* mutations and *FGFR1* amplification, had worse PFS despite receiving CDK4/6 inhibition [51]. Additionally, changes in ctDNA over time, such as an increase in the mutant allele fraction, have been associated with acquired resistance. One study identified detectable acquired RB1 mutations in ctDNA after exposure to CDK4/6 inhibitor [52]. These findings highlight the emergence of somatic RB1 mutations as a potential mechanism of resistance to CDK4/6 inhibitors. ctDNA analysis holds promise as a real-time monitoring tool for detecting resistance-associated alterations and guiding treatment decisions.

### 4.3. Imaging Techniques for Assessing Resistance

Positron emission tomography (PET) has been explored as a means of assessing resistance to CDK4/6 inhibitors. For example, PET imaging using radiotracers that target cell proliferation markers, such as 18F-fluorothymidine (FLT), has shown promise in monitoring treatment response and detecting early signs of resistance. Elmi et al. investigated the use of two PET proliferation tracers, [18F]FLT and [18F]ISO-1, for evaluating the response to a combination of palbociclib and fulvestrant in breast cancer cell lines and MCF7 tumor-bearing mice [53]. The findings showed that [18F]FLT is more sensitive to immediate changes in S-phase proliferation, while [18F]ISO-1 can assess delayed changes associated with cell-cycle arrest and the transition to G0 quiescence. These findings highlight the possible utility of imaging studies as biomarkers to indicate response and resistance to treatment.

The identification and validation of biomarkers associated with CDK4/6 inhibitor resistance are crucial for personalized treatment strategies. Multi-omics, ctDNA analysis, imaging techniques, and particularly a combination of multiple approaches can provide valuable tools for predicting and monitoring resistance. The extensive genetic heterogeneity found within human tumor samples leads to the emergence of subclones with varying biological capabilities, following the principle of ‘survival of the fittest’ [54]. CDK4/6 inhibitors targeting ER-positive breast cancer can face challenges with drug resistance due to the presence of ER-negative cell populations or adaptation processes driven by the cell plasticity of cancer cells. To minimize sampling bias and appropriately tailor therapeutic regimens, all available omics technologies should be used to evaluate tumor heterogeneity through the analysis of primary tumors, circulating tumor cells (CTCs), ctDNA, and, whenever possible, biopsies of secondary lesions. However, it is important to note that further research and validation are required to establish the clinical utility of these approaches.

## 5. Rational Combinations and Novel Therapeutic Strategies

Overcoming resistance to CDK4/6 inhibitors requires the development of novel therapeutic strategies that target resistance pathways, which can be identified through translational research and clinical trials. One such example is the use of immunotherapy in CDK4/6 inhibitor-resistant breast cancer, which holds promise for patients with limited treatment options.

### 5.1. Overcoming Resistance through Combination Therapies

Combination therapies that target multiple pathways involved in CDK4/6 inhibitor resistance have shown promise in preclinical and clinical studies (Table 2). For example, the combination of CDK4/6 inhibitors with PI3K/AKT/mTOR pathway inhibitors such as alpelisib in breast cancer cell lines and patient-derived models has demonstrated that alpelisib can overcome resistance mediated by PI3K pathway activation [29]. A phase Ib trial explored the combination of palbociclib, fulvestrant, and the PI3K inhibitor taselisib in 25 patients with *PIK3CA*-mutant HR-positive MBC. The triplet therapy demonstrated a 37.5% response rate [55].

HER-2 overexpression has also been linked to CDK4/6 inhibitor resistance in ER+ MBC. The combination of CDK 4/6 inhibitors with HER2-targeted therapies has shown synergistic effects in overcoming resistance in HER2-positive breast cancer models [56,57]. These novel combinations aim to simultaneously inhibit multiple pathways involved in resistance, thereby enhancing their therapeutic efficacy. The monarcHER trial compared the efficacy of abemaciclib plus trastuzumab with or without fulvestrant to standard-of-care chemotherapy plus trastuzumab in advanced HER2-positive breast cancer in patients who had received at least two previous therapies [58]. The combination of abemaciclib, fulvestrant, and trastuzumab showed a significant improvement in PFS compared to chemotherapy plus trastuzumab, indicating that this chemotherapy-free regimen could be a potential alternative treatment option for patients with HR-positive, HER2-positive advanced breast cancer.

FRFG alteration is another potential resistance mechanism that has been studied for combined therapeutic approaches. One study suggests that breast cancer cells with FGFR1 alterations that exhibit resistance to CDK4/6 inhibitors and fulvestrant can be overcome by combining the FGFR tyrosine kinase inhibitor (TKI) with CDK4/6 inhibitors and ER antagonists [34]. Additionally, a phase Ib trial evaluated the combination of the FGFR inhibitor erdafitinib with fulvestrant and palbociclib in patients with FGFR-amplified HR-positive MBC [59]. The treatment demonstrated clinical activity, particularly in patients with high levels of FGFR amplification, but erdafitinib-related side effects led to treatment discontinuation in several patients. These studies highlight the potential for combination therapies targeting the FGFR pathway as a mechanism to overcome CDK4/6 inhibitor resistance.

### 5.2. Targeting Alternative Pathways

Insights gained from translational research have identified specific alternative pathways that can be targeted to overcome CDK4/6 inhibitor resistance (Figure 2). For example, Chand et al. demonstrated cell growth inhibition and an anti-tumor effect in *CCNE1*-overexpressed breast cancer cells using INCB123667, a potent and selective small-molecular inhibitor of CDK2 that is currently in clinical development [60].

Targeting WEE1, a crucial regulator of the G2 checkpoint in endocrine and CDK4/6 inhibitor-resistant breast cancer cells, effectively reduces cell proliferation and induces G2/M arrest, apoptosis, and DNA double-stranded breaks, offering a promising therapeutic strategy for treatment-resistant HR+ breast cancer [61]. A phase II study evaluated the efficacy of adavosertib, a WEE1 inhibitor, in patients with *CCNE1*-amplified advanced refractory solid tumors. Adavosertib demonstrated promising clinical activity, with an objective response rate of 27% and a manageable toxicity profile, suggesting that further investigation of WEE1 inhibitors as a potential treatment option for *CCNE1*-amplified tumors, either alone or in combination with other therapies [62], is warranted.

### 5.3. Immunotherapy Combined with CDK4/6 Inhibitors

Immunotherapy has revolutionized cancer treatment. For instance, immune checkpoint inhibitors (ICIs), such as pembrolizumab, have shown clinical benefit in triple negative breast cancer. However, single agent pembrolizumab demonstrated only modest responses in HR+ MBC [63]. Recent research has highlighted the immune modulatory effects of CDK4/6 inhibitors and their combined efficacy when used with standard immunotherapy. Several studies have demonstrated that CDK4/6 inhibitors can enhance antitumor immunity by promoting interferon production, improving tumor antigen presentation, suppressing regulatory T cell proliferation, and increasing tumor infiltration and the activation of effector T cells, thus potentially enhancing the response to ICIs (Figure 2) [64,65]. A phase I/II trial investigated the safety and efficacy of combining palbociclib, pembrolizumab, and letrozole in women with HR+ MBC [66]. The combination therapy demonstrated good tolerability, with a complete response rate of 31% observed in patients receiving this regimen as a first-line treatment. Further trials are needed to explore the immune-priming effects of CDK4/6 inhibitors in HR+ MBC.

Rational combinations and novel therapeutic strategies offer potential new avenues to overcome CDK4/6 inhibitor resistance. The development of these approaches requires a comprehensive understanding of the underlying resistance mechanisms and their targeted modulation. Preclinical studies and ongoing clinical trials provide valuable insights into the efficacy and safety of these strategies in overcoming drug resistance and improving patient outcomes.

## 6. Future Directions and Conclusions

Despite significant progress, several questions regarding CDK4/6 inhibitor resistance remain unanswered. Further research is needed to elucidate the precise mechanisms underlying resistance development, including the interplay between different resistance pathways and the role of tumor heterogeneity in driving resistance. Additionally, the impact of the tumor microenvironment, including immune cells and stromal components, on CDK4/6 inhibitor response and resistance requires further exploration. Moreover, understanding the dynamics of resistance acquisition and the potential for the reversibility or persistence of resistance is crucial. Ongoing efforts in preclinical and clinical research aiming to address these questions will provide further insights into novel therapeutic targets and strategies.

## Figures and Tables

**Figure 1 ijms-24-11791-f001:**
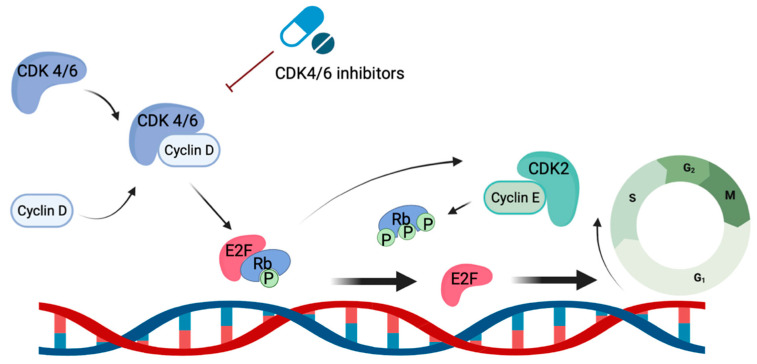
Signaling pathway of CDK 4/6-Cycle D complex and mechanism of action of CDK 4/6 inhibitors.

**Figure 2 ijms-24-11791-f002:**
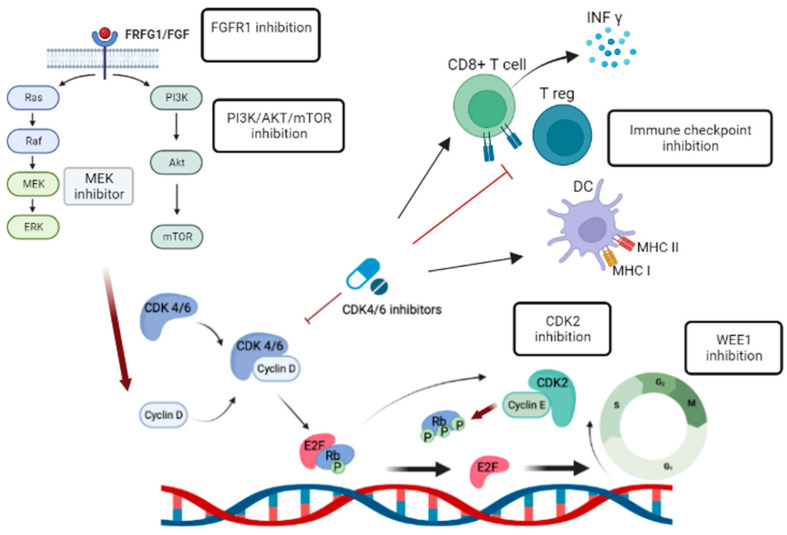
Targeting alterative pathway.

**Table 1 ijms-24-11791-t001:** Clinical trials and survival benefits of CDK4/6 inhibitors.

Endocrine-Sensitive MBC, Postmenopausal		N (Ratio)	mPFS	HR	mOS	HR
PALOMA-2 [4,5]	Palbociclib plus letrozole	666 (2:1)	24.8	0.58	53.9	0.956
	Placebo plus letrozole		14.5	1	51.2	1
MONALEESA-2 [6,7]	Ribociclib plus letrozole	668 (1:1)	25.3	0.56	63.9	0.76
	Placebo plus letrozole		16.0	1	51.3	1
MONARCH-3 [8]	Abemaciclib plus NSAI*	493 (2:1)	28.1	0.54	pending	
	Placebo plus NSAI*		14.7	1		
Endocrine resistant MBC, any menopausal status						
PALOMA-3 [9,10]	Palbociclib plus Fulvestrant	521 (2:1)	9.5	0.46	34.9	0.81
	Placebo plus Fulvestrant		4.6	1	28.0	1
MONARCH-2 [11,12]	Abemaciclib plus Fulvestrant	669 (2:1)	16.3	0.55	46.7	0.76
	Placebo plus Fulvestrant		9.3	1	37.3	1
Endocrine sensitive or resistant MBC, postmenopausal						
MONALEESA-3 [13,14]	Ribociclib plus Fulvestrant	726 (2:1)	20.5	0.59	NR	0.72
	Placebo plus Fulvestrant		12.8	1	40.0	1
Endocrine sensitive MBC, pre/perimenopausal						
MONALEESA-7 [15,16]	Ribociclib plus NSAI*/Goserelin	672 (2:1)	23.8	0.55	NR	0.72
	Placebo plus NSAI*/Goserelin		13.0	1	40.9	1

*NSAI: Non-steroidal aromatase inhibitor.

**Table 2 ijms-24-11791-t002:** Clinical trials of combination therapy with CDK 4/6 inhibitors.

Treatment Drug	Trial ID	Phase	Cancer Type
Ricociclib, everolimus, and exemestane	NCT02732119	1	HR+ HER2− locally advanced/metastatic breast cancer post progression on CDK 4/6 inhibitor
Palbociclib, bosutinib, and fuvestrant	NCT03854903	1	HR+ HER2− advanced breast cancer refractory to CDK 4/6 inhibitor
Palbociclib, letrozole, and venetoclax	NCT03900884	1b	ER and BCL2-positive breast cancer
Ribociclib, belinostat	NCT04315233	1/1b	Metastatic triple-negative breast cancer
Palbociclib and avelumab	NCT04360941	1b	AR+ triple-negative breast cancer
CDK4/6 inhibitor, fulvestrant, and capivasertib	NCT04862663	3	Metastatic HR+ HER2− breast cancer
CDK4/6 inhibitor, fulvestrant, and ipatasertib	NCT04920708	2	Metastatic HR+ HER2− breast cancer
Abemaciclib and tucidinostat	NCT05464173	1/2	Previously treated with palbociclib in HR+ HER2− relapsed/metastatic breast cancer
Ribociclib and alpelisib	NCT05508906	1b	Metastatic HR+ HER2− breast cancer

## Data Availability

Not applicable.

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
