# Peer review of "CDK4/6 Inhibitor Resistance in Hormone Receptor-Positive Metastatic Breast Cancer: Translational Research, Clinical Trials, and Future Directions"

_ijms, 2023, doi:10.3390/ijms241411791_

Round 1

Reviewer 1 Report

In this manuscript Lee et al. reviewed the research in the area metastatic breast cancer treatment with CDK4/CDK6 inhibitors. These inhibitors are widely used for treating hormone receptor positive breast cancer. The authors reviewed the mechanisms of resistance to therapy, which is often encountered. This is a fairly well-written review and I believe that the manuscript is suitable for publication in International Journal of Molecular Sciences. However, I have a minor point for the authors to consider. 

Minor point

There are some ER-positive breast cancers such as inflammatory breast cancer that respond poorly to CDK4/CDK6 inhibitors. It may be useful for the reader if resistant subpopulations are described in a bit more detail than what is in the manuscript. Could a high degree of intratumor heterogeneity and tumor adaptability be responsible? There is an increased realization that cancer evolves according to Darwinian principles, wherein cellular diversity and selection pressures shape the outcome. Unless therapeutic interventions consider this reality, therapies will have a limited impact particularly in highly heterogeneous breast cancers that do not respond to current therapies.

Author Response

Dear reviewer,

Thank you for your thoughtful comments. I deeply appreciate your insights and the critical attention you have given to the issue of tumor heterogeneity. I wholeheartedly agree with you on the significance of this topic.

Given the vast scope and complexity of tumor heterogeneity, it could indeed warrant a separate, comprehensive discussion on its own. However, considering the limitations of space in the manuscript, I opted to emphasize its importance by adding a dedicated paragraph in the lines 284-292: “The extensive genetic heterogeneity found within human tumor samples leads to the emergence of subclones with varying biological capabilities, following the principle of 'survival of the fittest.’ CDK4/6 inhibitors targeting ER positive breast cancer, can face challenges with drug resistance due to the presence of ER negative cell populations or adaptation processes driven by cell plasticity of cancer cells. To minimize sampling bias and appropriately tailor therapeutic regimens, all available omics technologies should be used to evaluate tumor heterogeneity through the analysis of primary tumors, circulating tumor cells (CTCs), ctDNA and, whenever possible, biopsies of secondary lesions”

Thank you for your review. 

Reviewer 2 Report

This review article is very well described, covering all the field of the resistance of CD4/6 inhibitors in breast cancer.  In particular, the detailed information on the mechanism of the resistance on the bases of clinical trials is very important contribution of this review article.  In addition, this article suggests new approaches for bypassing the resistance to the CD4/6 inhibitors.  Thus, this article is appropriate for the publication.

Author Response

Dear Reviewer,

I greatly appreciate your comments. 

Reviewer 3 Report

The authors of this review article layout a general overview of the available data regarding development of CDK4/6 inhibitors in metastatic breast cancer. Overall, the text is well-written and easy to comprehend by a non-expert reader. I would only like to bring attention to two minor points:

-      - It might be useful for the reader to introduce a small section (in the introduction) mentioning potential side effects of CDK4/6 inhibitors.

-    -   In lines 192-194 the authors must define whether PFS correlate with higher or lower PIK3CA changes.

Author Response

Dear reviewer,

Thank you for your thoughtful comments. In response to your feedback, we have carefully revised the manuscript, incorporating the suggested changes as outlined as below:

  1. Lines 60-64: We have included a section detailing the common adverse events associated with each medication.
  2. Lines 195-198: We have provided a clearer conclusion of the referred study, enhancing the overall coherence of our paper as “A separate investigation utilizing plasma samples from patients in the PALOMA-3 study revealed that early suppression of PIK3CA circulating tumor DNA (ctDNA) levels after 15 days of treatment of palbociclib and fulvestrant strongly correlated with improved PFS”

Thank you for your review.